# Age Readings and Assessment in Coastal Batoid Elasmobranchs from Small-Scale Size-Selective Fishery: The Importance of Data Comparability in Multi-Specific Assemblages

Umberto Scacco [1,2,*] , Fabiana Zanardi [2], Silvio Kroha [3] , Emanuele Mancini [4,5,6], Francesco Tiralongo [7,8] and Giuseppe Nascetti [2]

1    Italian Institute for Environmental Protection and Research, ISPRA (ISPRA), National Centre of Laboratories, Biology (CN-LAB-BIO), 00128 Rome, Italy
2    Department of Marine Eco-Biology (DEBM), University of Tuscia, 01100 Viterbo, Italy; fabiana.zanardi1994@gmail.com (F.Z.); nascetti@unitus.it (G.N.)
3    Department of Science, University of 'Rome 3', Viale Guglielmo Marconi, 00146 Rome, Italy; silvio.kroha@gmail.com
4    Department of Biological and Environmental Sciences and Technologies, DiSTeBA, University of Salento, 73100 Lecce, Italy; emanuele.mancini@unisalento.it
5    National Biodiversity Future Center (NBFC), 90100 Palermo, Italy
6    Ente Fauna Marina Mediterranea—Scientific Organization for Research and Conservation of Marine Biodiversity, 96012 Avola, Italy
7    Department of Biological, Geological and Environmental Sciences, University of Catania, 95124 Catania, Italy; francesco.tiralongo@unict.it
8    National Research Council, Institute of Marine Biological Resources and Biotechnologies, 60125 Ancona, Italy
*    Correspondence: umberto.scacco@isprambiente.it

**Abstract:** The large variation in vertebral shape and calcification observed among elasmobranch species prevents using a single method for enhancing growth bands and reading age. Further, estimating age and growth parameters can be difficult or impractical when samples are incomplete due to the bycatch of a size-selective fishery. Using a single and rapid method, age readings were obtained on the vertebrae of four batoid species, namely 53 individuals of *Dasyatis pastinaca*, 51 of *Raja asterias*, 15 of *Torpedo marmorata*, and 55 specimens of *Torpedo torpedo*, from the local small-scale trammel net fishery in the coastal waters (5–20 m depth) of the Central Tyrrhenian Sea during 2019–2021. Based on these data, a statistical routine was developed to obtain multiple estimates of age and growth parameters for incomplete samples due to size-selective fishing. The acceptable agreement between and within readers (intra and inter-reader disagreement < 5%) and the rate of increase in vertebral size with body size (differently ranked across species) demonstrated the consistency of the enhancing method. The parameters estimated by the Von Bertalanffy and Gompertz growth models matched the data available in the Mediterranean Sea for the species studied, with *D. pastinaca*, *T. torpedo*, and *R. asterias* showing the lowest (k = 0.05–0.12), intermediate (k = 0.112–0.19), and highest (k = 0.18–0.23) growth rates, respectively, in line with the life history traits of these species. Overall, the method proved effective both in delineating band pairs in vertebrae of different species and in reliably estimating the age and growth parameters of problematic samples due to size-selective fishing. The proposed method supports the collection of comparable demographic data from other areas where similar multi-specific assemblages are bycatch of size-selective fisheries impacting potential nursery areas and other essential habitats for elasmobranchs.

**Keywords:** aging methods; skates; rays coastal fishery; Mediterranean Sea

## 1. Introduction

Since 1980, several techniques and methods have been developed to read age on the inner vertebral surface of elasmobranch species [1–6]. Most aging techniques involve

staining procedures with expensive reagents (e.g., alizarin red, hematoxylin, xylene, crystal violet, graphite microtopography, cobalt nitrate, ammonium sulfide, Safranin O, Alcian blue, copper lead, iron-based salts, and silver nitrate), the operation of which depends strictly on the species being analyzed [4,7–12]. This is particularly true in the vertebrae of batoid species, as their vertebrae are difficult to read without histology or staining of some sort in many species [13]. On the other hand, the vertebrae of coastal species, such as carcharhinids and lamnids, do not require any stain techniques and can be read directly [13] using the most advanced spectroscopy methods, from near-infrared spectroscopy to X-ray techniques [4,8,14–16].

Vertebral morphology is critical in choosing the appropriate method, as shape and calcification can vary widely among species such that techniques for determining age in vertebral structures of elasmobranchs are generally species-specific [7,13,17–21]. For example, deep-water species have deep-coned, poorly calcified vertebrae, in contrast to the more calcified, less acute centra of coastal species [8,22,23]. While in the former case, the use of staining techniques is almost inevitable, in the latter case direct observations of growth increments are possible, along with the use of staining or other enhancement methods [6,8,23]. In this context, a multispecies method that avoids the staining step and uses portable stereoscopic equipment can be of value to speed up age readings and to obtain demographic information that is comparable across different species.

Estimating age and growth parameters can be difficult or impractical when estimating age in elasmobranch samples that are bycatch in a size-selective fishery [24]. On these occasions, samples are not complete in terms of both the number of observations and the representativeness of all expected age classes for the species under consideration [25]. There is a large amount of literature on VBGE fitting in different sample situations [24], as well as on the use of the increase in the size of biological structures, such as otoliths [26,27] and the vertebra in species of elasmobranchs to perform aging studies [4]. Since age–growth parameters are intrinsic characteristics of the sample [28], their estimation is optimal when it can be performed simultaneously on well-sized and length-structured samples [29]. Unfortunately, samples are generally far from optimal as they usually suffer from poorly represented size ranges and/or errors in age estimation [30], such as samples from size-selective fisheries. The small-scale fishery is one of the most size/species-selective fisheries, as it relies on the use of several highly selective passive fishing gears, such as fixed and drift nets, hooks, and pots [31–35]. This type of fishing is concentrated in coastal waters [36], sometimes very close to shore, as in the study area, as well as in other areas of the Italian and Mediterranean seas [37–39].

The multi-specific method developed was tested on four batoid elasmobranch species that are typical of inshore and even brackish waters on soft sandy–muddy bottoms of the Mediterranean Sea [40]. *Dasyatis pastinaca* (Linnaeus, 1758), *Raja asterias* Delaroche, 1809, *Torpedo marmorata* Risso, 1810, and *Torpedo torpedo* (Linnaeus, 1758) are emblematic of the bycatch of different trawling metiers in the Mediterranean Sea, such as bottom trawl [41] and small-scale fishery [37,42]. Although these species share a similar habitat, the biological diversity between them is notable. For instance, the common sting ray *D. pastinaca* is an opportunistic feeder [38,43,44] and an aplacental viviparous species [45] compared to the starry ray *R. asterias*, which is an oviparous [45] and coastal endemic species to the Mediterranean, particularly the western sector [46]. The opportunistic feeder starry ray [47] has some commercial importance, albeit low, in the central Tyrrhenian Sea, particularly in the landings of the trawl fishery [48]. The common torpedo *T. torpedo* and the electric marbled ray *T. marmorata* are sympatric species with slightly different bathymetric distribution and biological characteristics [49]. The common torpedo inhabits shallower depths than the electric marbled ray [50], but both species share a piscivorous feeding strategy [41,51,52] and aplacental viviparity [53,54]. The electric marbled ray inhabits a more heterogeneous habitat [55] and grows larger than the common torpedo, with a common length of about 60 cm [54] and up to 100 cm maximum total length recorded [56]. Age and growth studies in the Mediterranean Sea are generally scanty and/or dated for

the species here considered, such as works carried out for *D. pastinaca* in the Turkish [57], Greek [58], and Egyptian [59] waters; for *T. torpedo in* Spanish waters [60] and in the Gulf of Tunis [61]; for *R. asterias* in the northern Tyrrhenian Sea [62] and northwestern Mediterranean [41]; and for *T. marmorata* in Sardinian waters [63]. Based on these data, the common stingray and the starry skate exhibit the lowest and the highest growth rates, respectively, among the species considered, with the marbled electric ray having slower growth compared to the common torpedo.

The aim of this work is to provide age and growth estimates in a multi-specific assemblage of coastal batoid elasmobranchs by a unique and rapid method for reading and assessing age in problematic samples that are usually bycatch of size-selective coastal fisheries. The proposed method can help obtain reliable and comparable age and growth data for coastal elasmobranch species.

## 2. Materials and Methods

### 2.1. Sample Collection

All samples were collected through a fishery-dependent survey (on seven occasions between October 2019 and March 2021) planned at the Montalto di Castro (Central Tyrrhenian Sea, Italy) small-scale fisheries landing site in Fiora's Port Canal (Figure 1). The species sampled were from the bycatch of trammel nets used in coastal waters in the area (Figure 1) in a depth range of 5 to 20 m. The net was about 2 km long, with a mesh size of 3 cm for the inner panel and 10–15 cm for the outer panel. The trammel net was usually lowered at sunset and hauled at dawn. Fishermen were trained to facilitate the release of live specimens into the sea whenever possible and safe, and to collect all dead elasmobranch samples despite their size. After collection, specimens were immediately taken to the laboratory and stored at −20 °C until dissection.

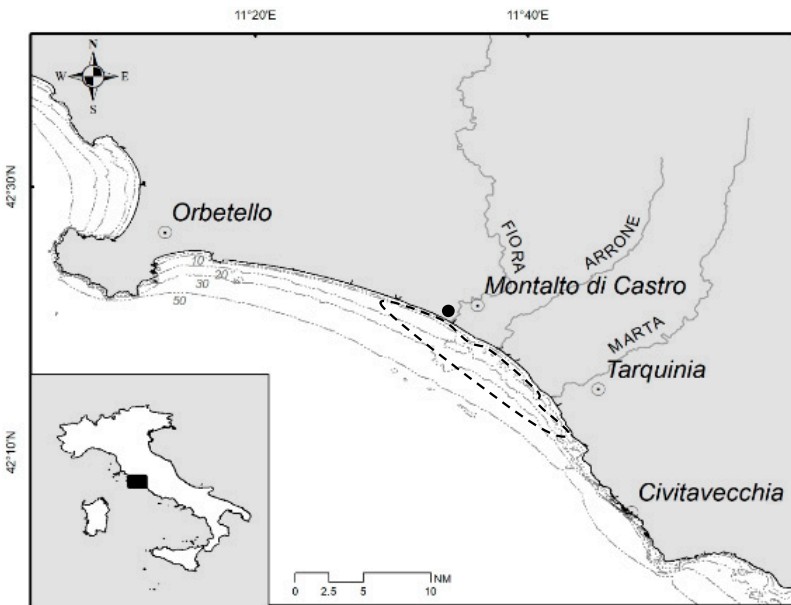

**Figure 1.** Map of the study area (central Tyrrhenian Sea, Italy, Mediterranean) with bathymetry, principal river mouths, and towns. The dotted figure encloses the area where samples came from and the dark circle indicates the local landing site chosen for sampling.

### 2.2. Biometrics

Specimens were identified taxonomically according to [40,64] and then measured and weighed to the nearest mm and mg for total length (TL) and disc width (DW) and weight, respectively. TL was not recorded for *D. pastinaca* because of the often-damaged tails. The specimens were sexed and the masses of liver, stomach, uterus, and dissected ovaries were weighed to the nearest mg. Sexual maturity was assessed on the scale developed by [65]. A

digital scale and a metal ichthyometer (scale sensitivity of 1 mg and 1 mm, respectively) were used to collect biometric data.

### 2.3. Laboratory Preparation of Vertebrae

A postcranial portion of the vertebral column approximately 5 cm long was removed from the animals, cutting away excess organic tissue as best we could. Based on a combined approach between vertebral preparations proposed by [5,21], several trials were performed to test the effectiveness of the method prior to reading and age assessment. Specifically, samples were tested to determine the bleach soak time necessary to remove organic tissue residues without compromising the growth increment reading. Once the optimal protocol was established, we followed the following steps:

1. The column section was immersed in NaClO (bleach) solution for between 30 and 60 min (larger sections took longer than smaller ones).

2. The column section was then rinsed with distilled water and sectioned into individual vertebrae.

3. The individual vertebrae were re-immersed in a new NaClO solution for 5–10 min depending on size, shaking them slowly to allow the solution to penetrate and remove residual tissue. Too short a time does not effectively remove all connective tissue, while too long a time literally destroys the vertebra.

4. Once removed from the solution, the vertebrae were rinsed with distilled water and allowed to dry under the laboratory fume hood.

5. For age determination, each vertebra was cut sagittally with a sharp scalpel. In this way, two halves (two half hourglasses for short) were obtained to observe the inner concave surface displaying the periodic increments arranged in series, from the vertebral center toward the margin. Next, the two halves were placed on a plate and immersed in simple glycerol to obtain synoptic observation of the symmetrical sides simultaneously. Next, the samples were observed with the Dino-Lite digital microscope (online information 1 a, b) in a dark room. The above transmitted LED light allowed us to observe the different contrast growth increments best visualized when hit by the multi-beam incident light oriented orthogonally of the sample. An external warm light lamp was used laterally to enhance the contrast of the bands.

6. Measurements of vertebral diameter (VD) and height (VH) of individuals were made using Dino-Lite digital microscope software on images transmitted directly to a personal computer with an accuracy of 0.001 mm (FIgures 2 and S1).

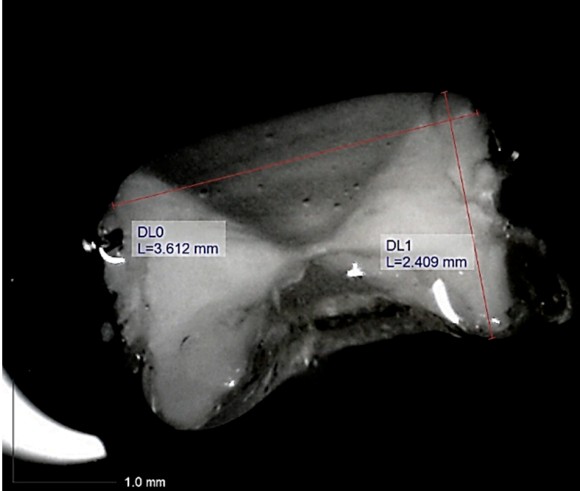

**Figure 2.** Example of the measurements (red perpendicular lines, L = size in mm; DL0 = vertebral diameter; DL1 = vertebral height) taken on a vertebral image obtained through observation under Dino-Lite equipment of an individual of *Dasyatis pastinaca* measuring 380 and 230 mm in total length and disk width, respectively (snapshot 1, acquisition date 15 June 2020).

7. Two readers independently performed age counts twice each on images selected in random order and without reference to specimen size for each species considered. One opaque and one translucent band was counted as a periodic increment. A band was considered distinct for counting when it was distinguishable from adjacent bands and observed continuously and correspondingly between the two sides of the sagittal section of the vertebra on its inner surface.

### 2.4. Statistical Methods

Intra- and inter-reader agreement on age estimates was checked by calculating the average percent error index (IAPE, [66]) on each species separately. VD and VH, as well as their ratio (VD/VH), were regressed on DW and TL using a linear model of the following type:

$$y = a \times x \pm b.$$

A test of parallelism was used in pairwise comparisons of the slopes of the regressions between species for each variable and predictor, separately. In the pairwise multiple comparisons, the p-level for acceptance of the results was lowered by Bonferroni's correction [67]. The normality of the size frequency distributions of the studied species was checked by Liliefors, Komolgorov–Smirnov, and Shapiro's tests. The difference between the sex ratios of the species was checked through separate chi-square tests. Because of the incomplete range of sample sizes (consisting mostly of juveniles and subadults), separate bootstrapping procedures of random resampling with replacement of individual length data (such as VH, VD, DW, and TL) were performed according to age in the original samples of each species. The number of iterations was set at 1000 to ensure a stabilized cumulative mean as close to the true sample mean as possible. The analytic geometric method of [68,69] replicated on each generated sample was used to obtain the overall mean asymptotic lengths provided with error (std. err, 95% conf. limits) over 1000 iterations. In this task, only models with 0 < slope < 1 and intercept > 0 were used. In fact, negative slopes equal to and greater than 1 imply a lack of fit to estimate asymptotic dimensions [68,69], as do distant models with intercept $\leq$ 0. Finally, these values were entered as fixed asymptotic parameters in the Von Bertalanffy logistic equations [70]

$$L\,(t) = L_\infty \times (1 - e\,\hat{}\,(-k \times (t - t_0)),$$

and in the Gompertz model [71]

$$L\,(t) = L_\infty \times e\,\hat{}\,(-e\,\hat{}\,(c_G - (k_G \times t)))$$

to fit the corresponding length-at-age data and calculate the parameters $k$, $t_0$, $k_G$, and $c_G$ with their error intervals, where

$\quad$ $k$ is the rate of growth in length with increasing age,
$\quad$ $t_0$ is the estimated time at birth,
$\quad$ $k_G$ is the Gomperz rate of growth, and
$\quad$ $c_G$ is the Gomperz parameter.

The estimation method and loss function were the Levenberg–Marquardt algorithm and least squares minimization for linear and nonlinear fit, respectively. Due to the limited sample size, unbalanced sex ratio, and uneven age distribution, growth parameters were not estimated for *T. marmorata*. Statistical analyses were performed with STATISTICA 7.1 [72], and Office Excel version 18.2110.13110.0 [73] was used to perform bootstrapping on the original samples by resampling with a replacement.

## 3. Results

### 3.1. Age Reading Agreement

The agreement between the two independent readings showed an error range that can be considered acceptable (<5%) for all studied species, both inter- and intra-reader (Table 1).

However, the IAPE was different among the species (Table 1). *T. torpedo* showed the lowest value, *R. asterias* and *T. marmorata* the intermediate values, and *D. pastinaca* the highest IAPE (Table 1). The distribution of the error also differed as the age of the species considered increased. The error occurred randomly in 12 specimens of *D. pastinaca* of different sizes, with a difference of one count between readers in all cases except two counts in the largest specimen (Figure A3a). *Raja asterias* showed a similar pattern (seven individuals), with a difference of one count in all cases (Figure A2b). In *T. torpedo*, the error was concentrated in the large individuals, but occurred in a few of them (3), with a difference of one count (Figure A2d). Similarly, the error increased with size in *T. marmorata*, with a two-count difference between readers in the largest specimen (Figure A2c).

**Table 1.** Index of Average Percent Error (IAPE) calculated between and within two readers based on two independent age readings within each reader. Data refer to a fishery-dependent assemblage sample of four coastal batoid species from the central Tyrrhenian Sea. Sample size by species is also provided.

| IAPE by Species | Sample Size | Intra-Reader 1 | Intra-Reader 2 | Inter-Readers |
|---|---|---|---|---|
| *Dasyatis pastinaca* | 53 | 3.33% | 2.91% | 4.26% |
| *Raja asterias* | 51 | 0.92% | 1.02% | 1.67% |
| *Torpedo marmorata* | 15 | 1.48% | 2.08% | 2.81% |
| *Torpedo torpedo* | 55 | 0.16% | 0.59% | 0.73% |

### 3.2. Photographic Evidence

Photographic images showed that the contrast and sharpness of the bands varied among species. Common stingrays had weaker bands (Figure 3a–c), particularly in older specimens (5–6 counts old) (Figure 3c), compared to the other species. *D. pastinaca* had poorly contrasted outer bands, with thin opaque areas and large intermediate areas that were poorly translucent. In the case of the starry skate (Figure 3d–f), the opaque and translucent zones had greater contrast than in the earlier species. However, in the oldest individual (Figure 3f) of the starry skate, some degree of folding of the outer margin was noted. Periodic increments were clear in the common torpedo, with good contrast and sharpness (Figure 3g–i). A similar situation was observed for the marbled electric ray (Figure 3j–l), with darker opaque bands in young and intermediate specimens (Figure 3j,k) than in older ones. Birthmark was observed particularly in the sectioned vertebrae of *T. marmorata* and *R. asterias*, as a change in angle of the corpus calcareum was noted in small (Figures 3d and 3j, respectively), intermediate (Figures 3e and 3k, respectively), and large individuals (Figures 3f and 3l, respectively) of these species.

### 3.3. Regression Results and Characteristics of the Samples

The goodness of fit of the linear model varied across species and variables (Table 2, Appendix A). However, the relationships showed a general linear increase in diameter and vertebral height with increasing DW (Figure 4a–d) and TL (Figure A1a–c) in all species, although this increase was not significant in *T. marmorata* (Table 2, Appendix A). The common torpedo showed the steepest slopes for both VD and VH along DW and TL (Figures 4c and A1c; Table 2, Appendix A). In the other cases, regression slopes varied according to species, vertebral size, and predictor. For example, *R. asterias* (Figure A1a) and *D. pastinaca* (Figure 4a) showed the lowest slopes for the VD-TL and VH-DW regressions, respectively (Table 2, Appendix A). The VD/VH ratio followed an invariant trend with increasing size, or slightly decreasing in some species (Figures 4a–d and A1a–c). For this ratio, explained variances were very low and slopes were never significant in all species except *D. pastinaca* (Figure 4a, Table 2, Appendix A).

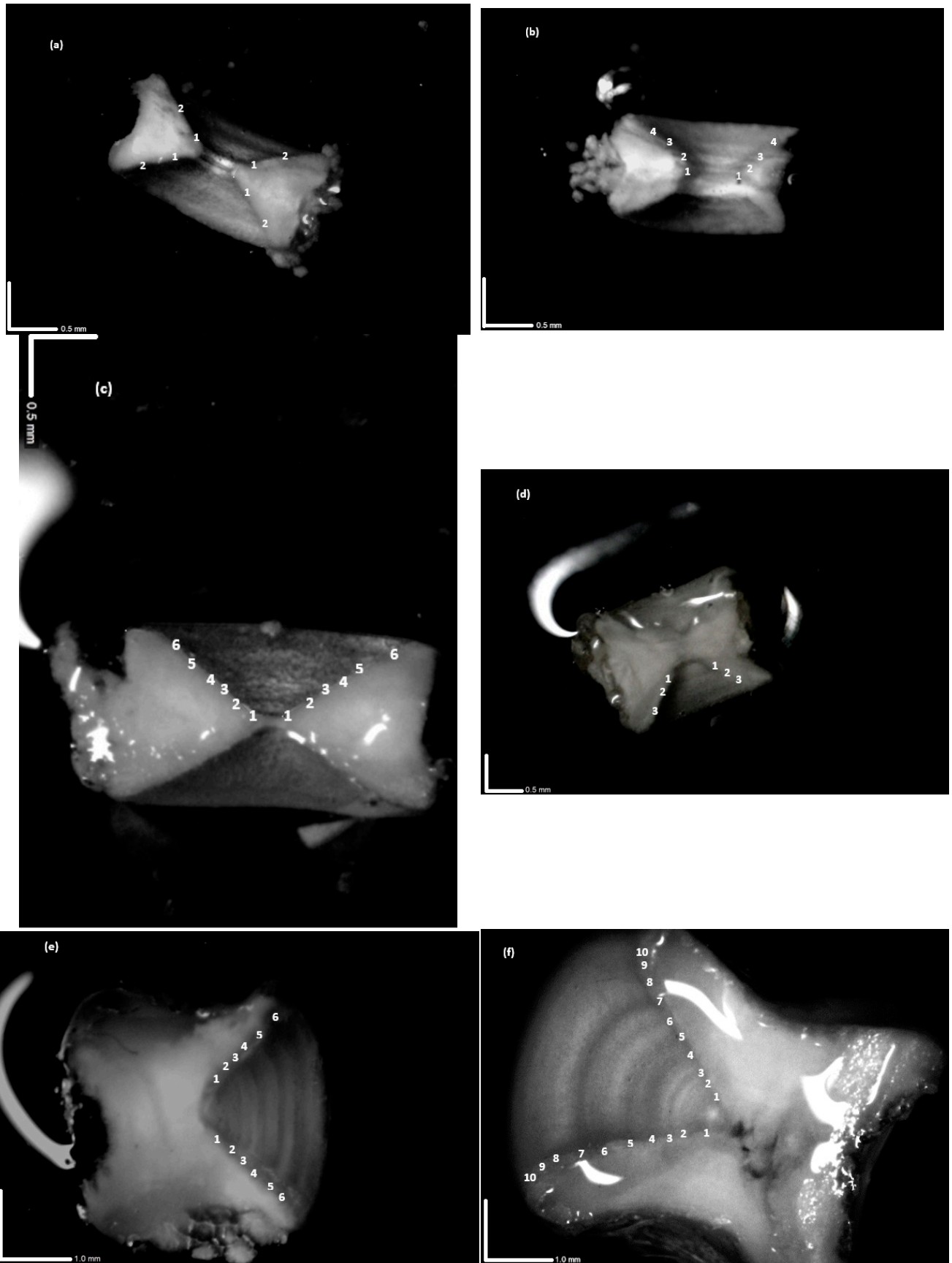

**Figure 3.** *Cont.*

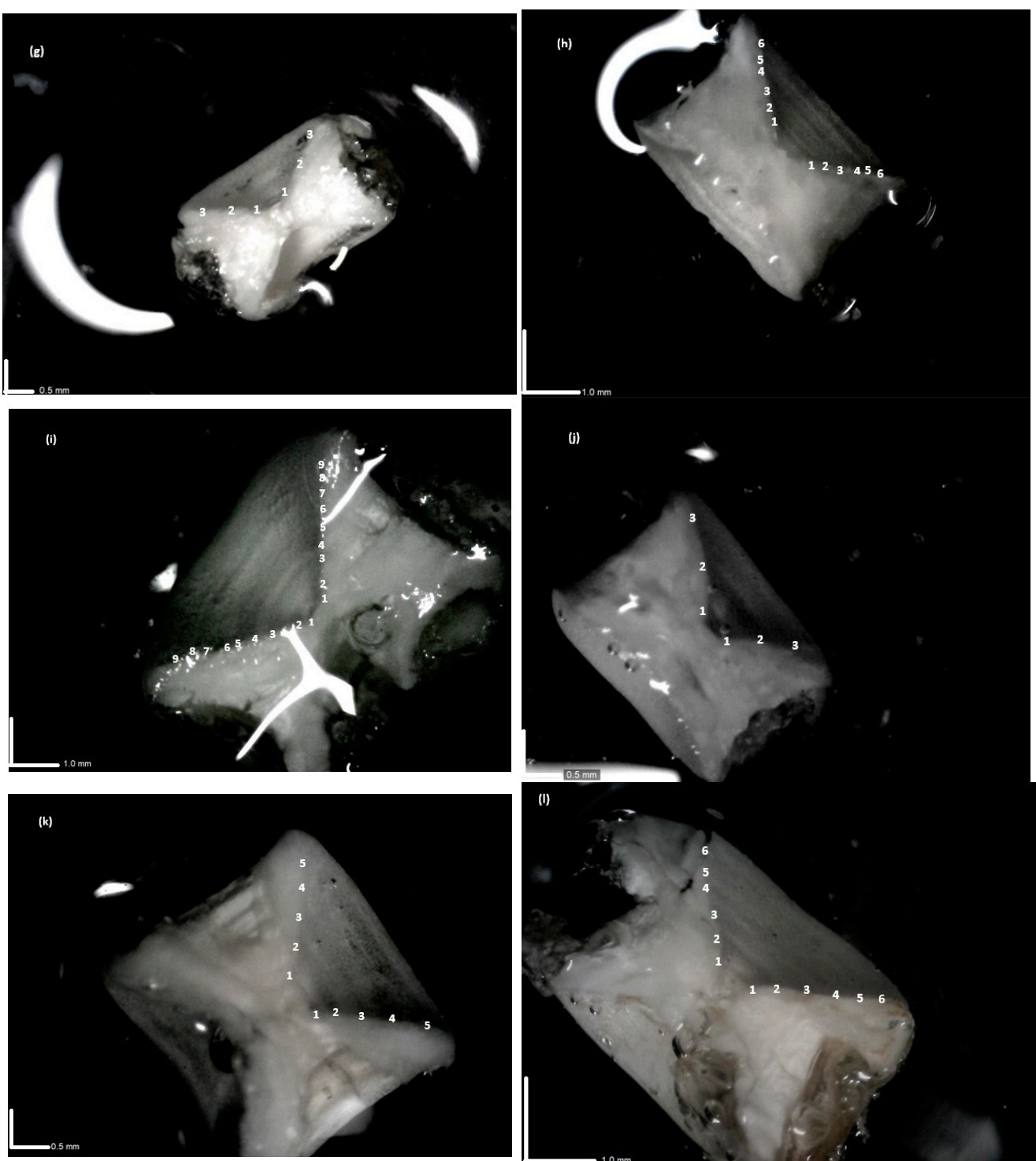

**Figure 3.** (**a**–**l**) Sagittal sections of vertebrae of four coastal batoid elasmobranch species observed in glycerol immersion under incident LED light by Dino-Lite equipment with indication of rings and corresponding age counts. *Dasyatis pastinaca* (**a**–**c**), *Raja asterias* (**d**–**f**), *Torpedo torpedo* (**g**–**i**), and *Torpedo marmorata* (**j**–**l**). Bold white scale bars are reported. Juveniles (**a**,**d**,**g**,**j**), intermediate (**b**,**e**,**h**,**k**), and large (**c**,**f**,**i**,**l**) individuals are shown. Magnification: (**a**) = 79.4 x, (**b**) = 81.7 x, (**c**) = 62.3 x, (**d**) = 59.9 x, (**e**) = 55.4 x, (**f**) = 55.4 x, (**g**) = 61.5 x, (**h**) = 43.5 x, (**i**) = 40.2 x, (**j**) = 62.3 x, (**k**) = 58.6 x, (**l**) = 51.9 x. Numbers indicate age counts.

**Table 2.** Regression parameters (RP, a: slope, b: intercept, R2: explained variance, Std. err: standard error; t (n − 2): degree of freedom of t statistic; and p: level of significance) of linear relationships between vertebral dimensions and increasing fish size as disk width (mm) of four coastal batoid species (DP: *Dasyatis pastinaca*, RA: *Raja asterias*, TM: *Torpedo marmorata*, and TT: *Torpedo torpedo*) sampled in the central Tyrrhenian Sea. VD, VH, and VD/VH are vertebral diameter, height, and their ratio, respectively. The same letter in brackets indicates a significantly different regression slope between species with asterisks denoting its p-level of significance. ns stands for not significant differences. * $p < 0.05$, ** $p < 0.01$, *** $p < 0.001$.

| Species | RP | DP | RA | TM | TT |
|---|---|---|---|---|---|
| VD | *a* | $1.2 \times 10^{-2}$ | $9.7 \times 10^{-3}$ | $8.0 \times 10^{-3}$ | $2.2 \times 10^{-2}$ |
| | *Std. err* | $1.1 \times 10^{-3}$ | $8.3 \times 10^{-4}$ | $5.6 \times 10^{-3}$ | $1.3 \times 10^{-3}$ |
| | *t (n − 2)* | 11.0 | 11.6 | 1.4 | 17.4 |
| | *p* | *** (b*) | *** (a **, b *) | ns | *** (a **) |
| | *b* | −0.23 | 0.61 | 2.17 | −0.38 |
| | *Std. err* | 0.23 | 0.23 | 1.0 | 0.23 |
| | *t (n − 2)* | −1.0 | 2.7 | 2.1 | −1.6 |
| | *p* | ns | * | ns | ns |
| | $R^2$ | 0.71 | 0.72 | 0.14 | 0.85 |
| VH | *a* | $9.3 \times 10^{-3}$ | $1.0 \times 10^{-2}$ | $1.1 \times 10^{-2}$ | $1.6 \times 10^{-2}$ |
| | *Std. err* | $1.1 \times 10^{-3}$ | $9.0 \times 10^{-4}$ | $5.8 \times 10^{-3}$ | $9.7 \times 10^{-4}$ |
| | *t (n − 2)* | 8.7 | 11.0 | 1.8 | 16.1 |
| | *p* | *** (c **, d *) | *** (d *) | ns | *** (c **) |
| | *b* | −0.48 | $1.0 \times 10^{-1}$ | 0.77 | −0.29 |
| | *Std. err* | $2 \times 10^{-1}$ | $2.4 \times 10^{-1}$ | 1.1 | $1.8 \times 10^{-1}$ |
| | *t (n − 2)* | −2.1 | 0.41 | 0.73 | −1.6 |
| | *p* | * | ns | ns | * |
| | $R^2$ | 0.61 | 0.70 | 0.21 | 0.83 |
| VD/VH | *a* | $-1.4 \times 10^{-3}$ | $-2.0 \times 10^{-4}$ | $-2.3 \times 10^{-3}$ | $1.1 \times 10^{-5}$ |
| | *Std. err* | $5.7 \times 10^{-4}$ | $4.1 \times 10^{-4}$ | $2.6 \times 10^{-3}$ | $4.3 \times 10^{-4}$ |
| | *t (n − 2)* | −2.4 | −0.6 | −0.9 | −0.9 |
| | *p* | * | ns | ns | ns |
| | *b* | 1.91 | 1.23 | 1.78 | 1.42 |
| | *Std. err* | $1.2 \times 10^{-1}$ | $1.1 \times 10^{-1}$ | $4.8 \times 10^{-1}$ | $2.5 \times 10^{-2}$ |
| | *t (n − 2)* | 15.7 | 11.1 | −3.7 | $1.0 \times 10^{-1}$ |
| | *p* | *** | *** | ** | *** |
| | $R^2$ | 0.11 | $6.0 \times 10^{-3}$ | $5.6 \times 10^{-2}$ | $1.2 \times 10^{-5}$ |

The size frequency distributions (Figure 5a–d) of the studied species had different deviances from normality. In the case of *D. pastinaca* (Figure 5a), distribution was right-skewed (asymmetry: 1.30 ± 0.33 std. err.) with respect to the sample mean DW (206.78 mm ± 6.32 std. err.) and not normal values (K-S: d = 0.25, $p < 0.01$; Lilliefors, $p < 0.01$; W Shapiro–Wilk test, W = 0.84, $p < 0.001$). Three different cohorts may be argued in this species, with observations more concentrated in the smaller compared to larger size class (Figure 5a). Differently, the distributions for *R. asterias* (K-S, d = 0.13, $p > 0.05$; Lilliefors, $p < 0.05$; Shapiro–Wilk test, W = 0.92, $p < 0.01$) and *T. marmorata* (K-S, d = 0.22, $p > 0.05$; Lilliefors, $p < 0.05$; Shapiro–Wilk test, W = 0.86, $p < 0.05$) appeared closer to normality compared to *D. pastinaca*. In fact, a moderate right (0.46 ± 0.33 std. err.) and left skewed asymmetry (−1.53 ± 0.58) was found with respect to the corresponding sample means (268.26 mm ± 6.48 std. err. and 181.27 mm ± 5.61 std. err.) in the distributions of *R. asterias* and *T. marmorata*, respectively. Therefore, observations were more concentrated in the larger and smaller size classes in *R. asterias* and *T. marmorata*, respectively (Figure 5b,c). The distribution was normal in the case of *T. torpedo* (K-S d = 0.05, $p > 0.05$; Lilliefors, $p > 0.05$; Shapiro–Wilk test, W = 0.96, $p > 0.05$), with negligible right skewness (asymmetry 0.32 ± 0.41 std. err.) with respect to the mean of the sample (179.82 mm ± 5.86 std. err.). In this case, the observations appeared to be roughly equally distributed around the mean

(Figure 5d). The sex ratios were balanced in *D. pastinaca* (F/M = 0.83, X2 = 0.49, d.f. = 1, *p* > 0.05) and *R. asterias* (F/M = 0.96, X2 = 0.02, df = 1, *p* > 0.05), while males were exclusive in *T. marmorata* and females prevailed over males in *T. torpedo* (F/M = 2.05, X2 = 6.56, d.f. = 1, *p* < 0.01).

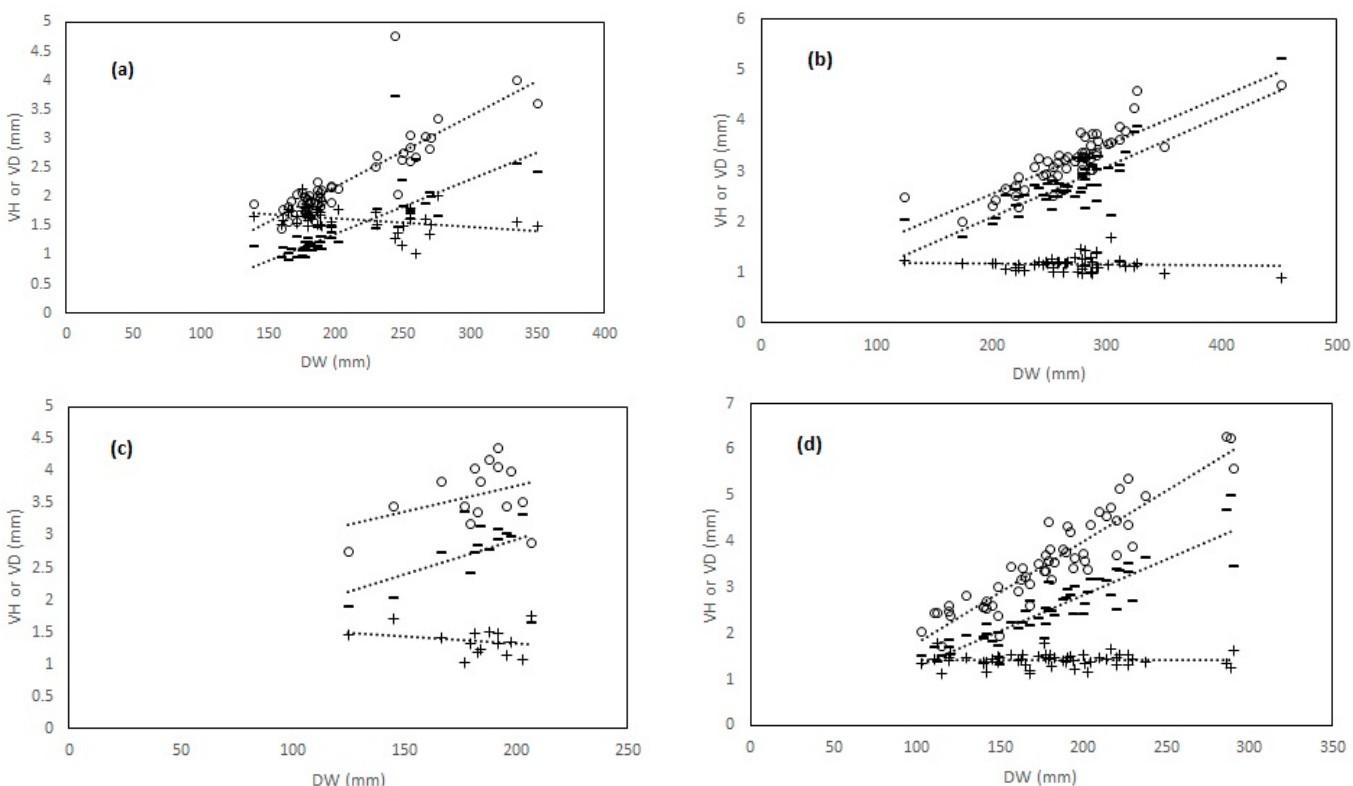

**Figure 4.** (**a**–**d**) Linear relationships (dotted lines) between disk width (DW) and vertebral measures (VD vertebral diameter, empty circles; VH vertebral height, minus; and VD/VH ratio, dark crosses) in four coastal batoid elasmobranch species form the central Tyrrhenian Sea. (**a**): *Dasyatis pastinaca*; (**b**): *Raja asterias*; (**c**): *Torpedo marmorata*; and (**d**): *Torpedo torpedo*.

*3.4. Bootstrapping and Age and Growth Parameters Estimation*

Out of a total of 1000 iterations performed for all species and variables, the cumulative sample averages reached stability after a different number of cumulative iterations, depending on the species and variable involved (Appendices D and E). For example, the number of iterations needed to stabilize the sample mean of *T. torpedo* was generally higher for all variables than for the other species (Appendices D and E). In contrast, the cumulative sample averages of VD and VH in *T. marmorata* remained unstable around and beyond the maximum number of iterations performed, respectively (Figure A3d).

The number of appropriate Ford–Walford fits (0 < slope < 1 and intercept > 0) to compute the mean asymptotic lengths differed depending on the species and variable in question, but with no apparent ordering along variables and/or species (Table 3). In terms of the error of the mean asymptotic estimates, it was acceptably narrow in all species for all variables except in *D. pastinaca* for VH∞ and DW∞. *Torpedo marmorata* showed lower errors along all variables than the other species (Table 4). Nevertheless, the estimates of asymptotic lengths were not realistic for this species (Table 3). Therefore, we excluded *T. marmorata* from further analysis.

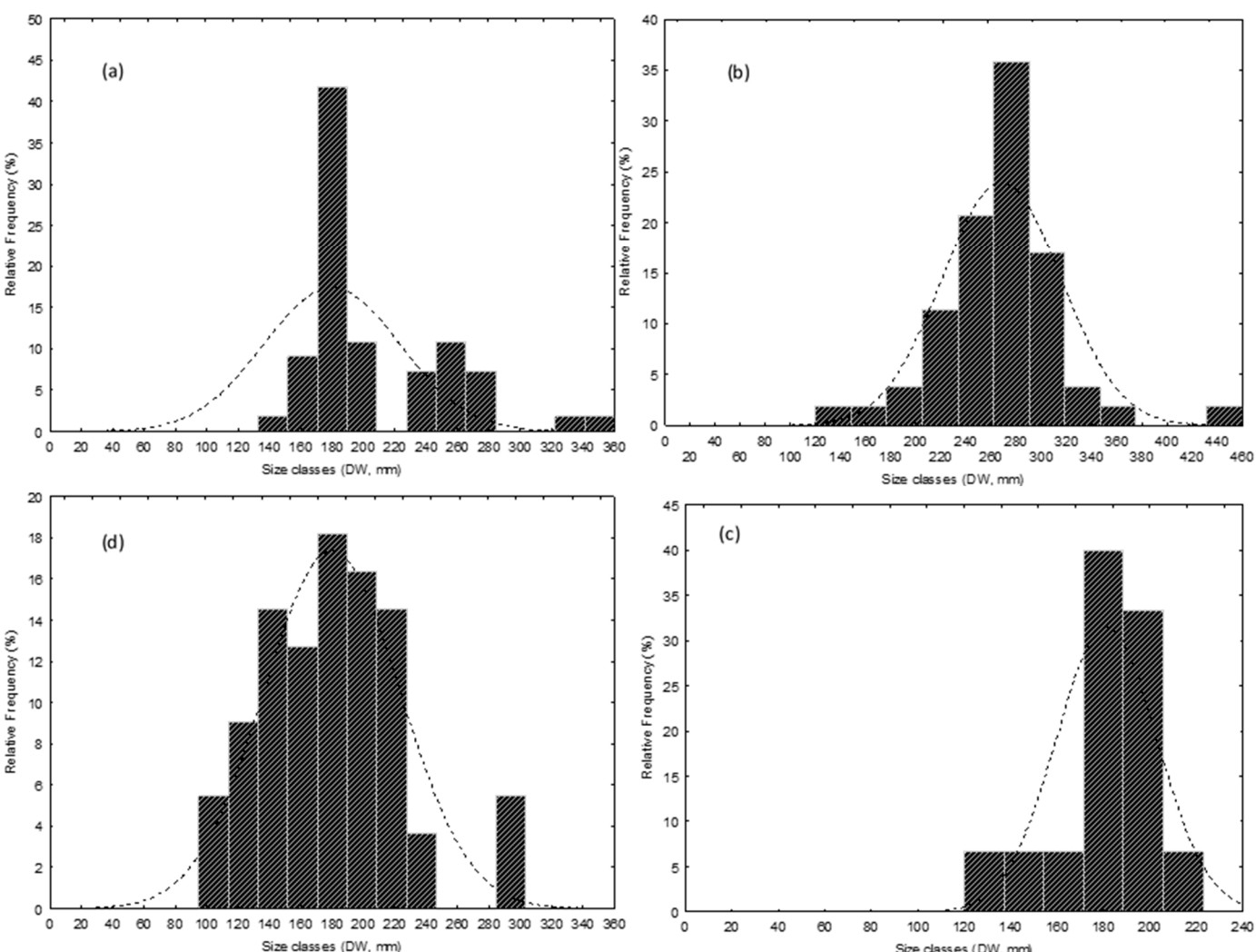

**Figure 5.** (**a–d**) Size-frequency distributions of four coastal batoid elasmobranch species from the bycatch of a size-selective small-scale fishery from the central Tyrrhenian sea. (**a**): *Dasyatis pastinaca*; (**b**): *Raja asterias*; (**c**): *Torpedo marmorata*; and (**d**): *Torpedo torpedo*. DW is disk width in mm.

On the basis of the results of both VBGE and Gompertz models, the mean explained variance of fits was generally lower when vertebral lengths were used as fixed asymptotic parameters rather than body lengths (Table 4). This was observed in all species except *T. torpedo*, which showed high $R^2$ despite the model and variable used (Table 4). In general, the Gompertz model produced better (negative) estimates of $t_0$ than the VBGE models for all species and variables (Figures 6 and 7a–d; Table 4). In fact, VBGE models produced positive $t_0$ estimates when vertebral measures VH and VD were used as asymptotic lengths in the models of *R. asterias* and *T. torpedo* (Figure 6a) and *T. torpedo* (Figure 6b), respectively (Table 4). The growth rates of species ($k_G$) were higher in the Gompertz model compared to the VBGE models (k) across all variables (Figures 6 and 7a–d; Table 4). In general, the range of growth rates estimated had the highest values in *R. asterias* (k and $k_G$ = 0.18 and 0.23, respectively), the intermediate ones in *T. torpedo* (0.12–0.19), and the lowest in *D. pastinaca* (0.05–0.12), comparing all variables and models across species (Figures 6 and 7a–d, respectively; Table 4).

**Table 3.** Errors (standard error and 95% upper and lower confidence limits) of mean estimated asymptotic lengths expressed as vertebral height (VH), diameter (VD), total body length (TL), and disk width (DW) and estimated through Ford–Walford fits over 1000 bootstrapping iterations (NISCM: approximate Number of Iterations needed to Stabilize Cumulated Mean length; NFWF: Number of Ford–Walford fits used to calculate mean asymptotic length) on original data of four coastal batoid elasmobranchs from the central Tyrrhenian Sea.

| Mean Asymptotic Length (mm) | | *D. pastinaca* | *T. torpedo* | *R. asterias* | *T. marmorata* |
|---|---|---|---|---|---|
| $VH_\infty$ | Est | 9.09 | 5.08 | 3.43 | 3.27 |
| | Std. err | ±3.50 | ±0.01 | ±0.06 | ±0.05 |
| | C. lim. −95% | 2.21 | 4.71 | 3.31 | 3.18 |
| | C. lim. +95% | 15.47 | 5.46 | 3.56 | 3.36 |
| | NISCM | 700 | 900 | 450 | >1000 |
| | NFWF | 474 | 606 | 561 | 338 |
| $VD_\infty$ | Est | 12.86 | 10.35 | 7.86 | 3.88 |
| | Std. err | ±0.01 | ±0.38 | ±0.24 | ±0.03 |
| | C. lim. −95% | 10.50 | 9.60 | 7.39 | 3.83 |
| | C. lim. +95% | 15.21 | 11.11 | 8.32 | 3.94 |
| | NISCM | 800 | 900 | 850 | ≈1000 |
| | NFWF | 152 | 491 | 542 | 279 |
| $TL_\infty$ | Est | na | 671.90 | 699.66 | 342.00 |
| | Std. err | | ±26.23 | ±21.26 | ±11.15 |
| | C. lim. −95% | | 620.36 | 657.25 | 320.12 |
| | C. lim. +95% | | 723.43 | 742.07 | 363.89 |
| | NISCM | | 850 | 600 | 900 |
| | NFWF | | 551 | 367 | 749 |
| $DW_\infty$ | Est | 1056.65 | 439.87 | 427.92 | 206.15 |
| | Std. err | ±372.81 | ±13.85 | ±34.76 | ±1.25 |
| | C. lim. −95% | 307.46 | 412.67 | 359.33 | 203.70 |
| | C. lim. +95% | 1805.85 | 467.07 | 496.52 | 208.61 |
| | NISCM | 450 | 950 | 800 | 950 |
| | NFWF | 50 | 638 | 177 | 767 |

**Table 4.** Errors (±Std. err.), estimated growth parameters (k and $t_0$: VBGE rate of growth and time at size 0, respectively; $k_G$ and $c_G$: Gompertz rate of growth and parameter, respectively), and statistics ($R^2$ and $t_{n-2}$: explained variance and t statistic for n − 2 degrees of freedom with n as sample size, respectively) of VBGE and Gompertz models calculated by imposing mean bootstrapped asymptotic lengths (values right to numbers in superscript brackets indicating correspondence with species) as vertebral height (VH∞), diameter (VD∞), total length (TL∞), and disk width (DW∞) for three species of coastal batoid elasmobranchs from the central Tyrrhenian Sea. * $p < 0.05$, ** $p < 0.01$, *** $p < 0.001$. Borderline p-levels are in brackets. ns: not significant.

| | | [1] *D. pastinaca* | | | | [2] *T. torpedo* | | | | [3] *R. asterias* | | | |
|---|---|---|---|---|---|---|---|---|---|---|---|---|---|
| | | VBGE | | GOMPERTZ | | VBGE | | GOMPERTZ | | VBGE | | GOMPERTZ | |
| | | k | $t_0$ | $k_G$ | $c_G$ | k | $t_0$ | $k_G$ | $c_G$ | k | $t_0$ | $k_G$ | $c_G$ |
| $VH_\infty$ [1] 9.09 [2] 5.08 [3] 3.43 | Est | 0.05 | −0.15 | 0.14 | −1.08 | 0.18 | 0.77 | 0.27 | −0.88 | 0.38 | 0.20 | 0.43 | −0.44 |
| | ±Std. err. | 0.01 | 0.60 | 0.08 | 0.02 | 0.02 | 0.30 | 0.02 | 0.10 | 0.09 | 0.93 | 0.11 | 0.43 |
| | $t_{n-2}$ | −5.85 | 0.25 | 12.92 | 6.05 | 11.46 | 2.54 | 11.84 | 8.61 | 4.06 | 0.22 | 4.07 | 1.02 |
| | p | *** | ns | *** | *** | *** | * | *** | *** | *** | ns | *** | ns |
| | $R^2$ | 0.42 | | 0.42 | | 0.77 | | 0.78 | | 0.40 | | 0.41 | |
| $VD_\infty$ [1] 12.86 [2] 10.35 [3] 7.86 | Est | 0.05 | −0.77 | 0.13 | −0.98 | 0.09 | 0.09 | 0.16 | −0.83 | 0.06 | −3.49 | 0.10 | −0.37 |
| | St.d err | 0.01 | 0.59 | 0.02 | 0.06 | 0.01 | 0.36 | 0.01 | 0.06 | 0.01 | 1.10 | 0.01 | 0.06 |
| | $t_{n-2}$ | 7.05 | −1.30 | 16.13 | 7.56 | 12.24 | 0.25 | 12.60 | 13.10 | 7.36 | −3.16 | 7.64 | 5.76 |
| | p | *** | ns | *** | *** | *** | ns | *** | *** | *** | ** | *** | *** |
| | $R^2$ | 0.51 | | 0.53 | | 0.76 | | 0.77 | | 0.54 | | 0.55 | |

**Table 4.** *Cont.*

| | | (1) *D. pastinaca* | | | | (2) *T. torpedo* | | | | (3) *R. asterias* | | | |
|---|---|---|---|---|---|---|---|---|---|---|---|---|---|
| | | **VBGE** | | **GOMPERTZ** | | **VBGE** | | **GOMPERTZ** | | **VBGE** | | **GOMPERTZ** | |
| $TL_\infty$ (2) 671.9 (3) 699.7 | Est | | na | | | 0.11 | −0.26 | 0.18 | −0.68 | 0.12 | −2.19 | 0.17 | −0.19 |
| | St.d err | | | | | 0.01 | 0.39 | 0.01 | 0.07 | 0.01 | 0.81 | 0.02 | 0.09 |
| | $t_{n-2}$ | | | | | 12.05 | −0.67 | 12.27 | 9.90 | 8.23 | −2.70 | 8.49 | 2.11 |
| | p | | | | | *** | ns | *** | *** | *** | ** | *** | * |
| | $R^2$ | | | | | | 0.76 | | 0.77 | | 0.63 | | 0.64 |
| $DW_\infty$ (1) 1056.6 (2) 439 (3) 427.9 | Est | 0.05 | −1.23 | 0.12 | −0.90 | 0.10 | −0.57 | 0.16 | −0.64 | 0.17 | −1.09 | 0.22 | −0.27 |
| | St.d err | 0.004 | 0.44 | 0.01 | 0.04 | 0.01 | 0.44 | 0.01 | 0.07 | 0.02 | 0.78 | 0.03 | 0.14 |
| | $t_{n-2}$ | 10.36 | .2.77 | 11.46 | 22.63 | 11.40 | −1.29 | 11.60 | 9.62 | 7.00 | −1.40 | 7.12 | 1.91 |
| | p | *** | ** | *** | *** | *** | ns | *** | *** | *** | ns | *** | (0.06) |
| | $R^2$ | | 0.70 | | 0.72 | | 0.74 | | 0.74 | | 0.57 | | 0.58 |

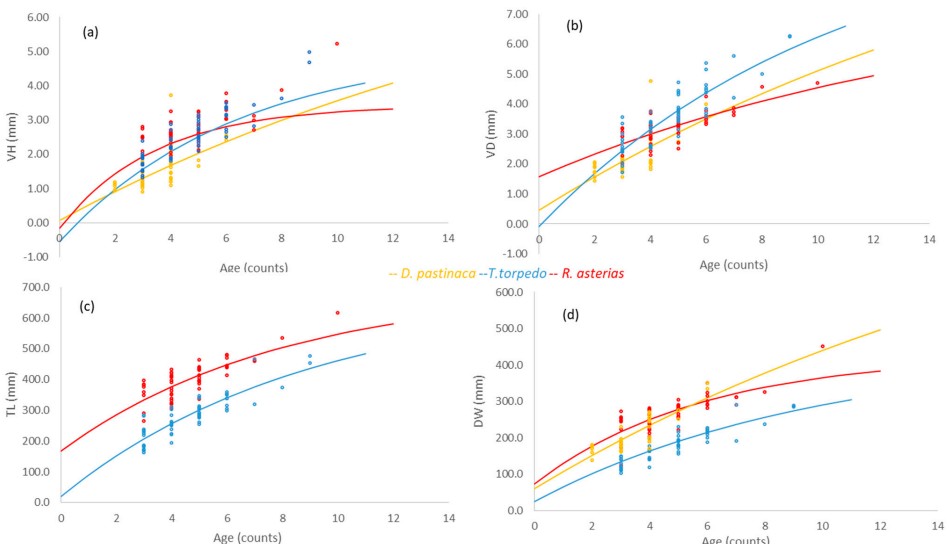

**Figure 6.** (**a**–**d**) VBGE models based on vertebral (**a**) height (VH) and (**b**) diameter (VD), (**c**) total length (TL), and (**d**) disk width (DW) used as asymptotic lengths in *Dasyatis pastinaca*, *Raja asterias*, and *Torpedo torpedo* from the central Tyrrhenian Sea.

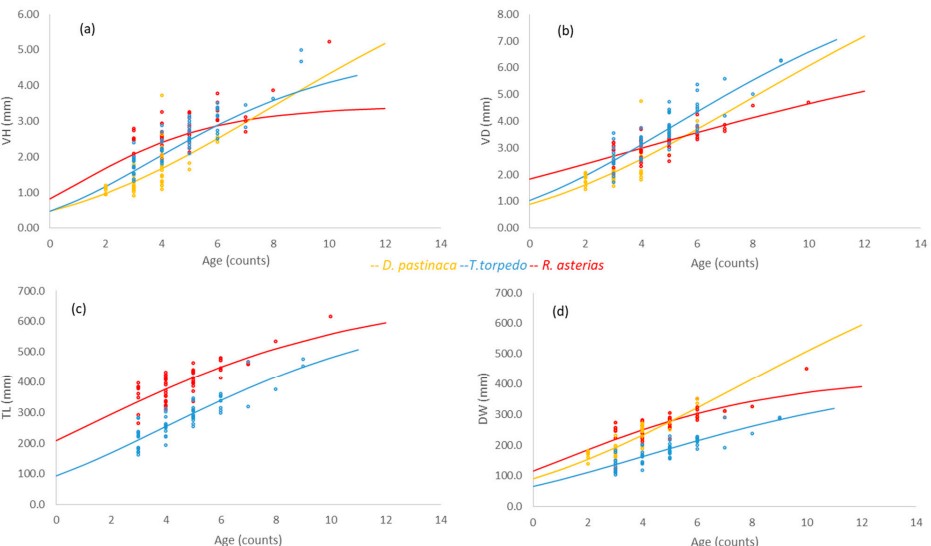

**Figure 7.** (**a**–**d**) Gompertz models based on vertebral (**a**) height (VH) and (**b**) diameter (VD), (**c**) total length (TL), and (**d**) disk width (DW) as asymptotic lengths in *Dasyatis pastinaca*, *Raja asterias,* and *Torpedo torpedo* from the central Tyrrhenian Sea.

## 4. Discussion

The study of demographic status and population trajectory of species under fishing pressure requires reliable age determination techniques [8], preferably rapid and inexpensive, as well as being multi-specific compared to those available. The method here presented allowed rapid and reliable reading of the vertebral age rings in four coastal batoid elasmobranchs. The use of an inexpensive and handy stereomicroscope using LED lights enhanced the action of multiple beams striking the specimen orthogonally. This enhanced the contrast of growth increments on vertebral sections simply immersed in glycerol as the only reagent. Another equally important achievement is the effectiveness of the statistical method developed to calculate age and growth parameters on size-selective and multi-specific fishery samples. Overall, the use of this method can be a great advantage, both in terms of cost-effectiveness and time taken to read age rings in different species and in terms of feasibility in experimental field settings, above all to provide comparable demographic information.

Verification of the method came from the consistency of the data we found at three different levels.

First, the good general agreement among observers demonstrated the internal consistency of age readings within and between them, i.e., the measurements can be considered repeatable [66]. However, different errors have occurred among species, which were generally concentrated in large individuals. This is expected in aging studies because of error propagation with increasing age [4] and morphological differences between the vertebrae of elasmobranch species [74,75]. The latter is the case of *D. pastinaca*, which showed the greatest error in age counting on its poorly calcified vertebrae, as well as more acute in shape than the other species, as indicated by the VD/VH ratio. Like the deep-coned vertebrae of deep-sea shark species [21], more concave, poorly calcified vertebrae are more difficult to read than the less acute ones [8,19–21,75] due to the concealing effect of growth bands and the folding of the outer vertebral margin [4]. The latter, which can occur also in batoid species, can add a significant age-count bias in the oldest specimens such that histological staining preparations are required in these samples [13].

Second, it was found that as body size, and therefore age, increased, vertebral size (height and diameter) increased in all species as well, as recommended for data verification in aging studies [75]. On the one hand, high age–growth rates did not correspond to high vertebral growth rates with increasing body size in the species studied. Indeed, the starry skate showed an intermediate to low rate of increase in vertebral size with increasing body size compared to the common torpedo. This may indicate different allometric growth of body parts between different species [13]. For example, skates have a wider "non-vertebral" portion of the body than torpedoes, as the former has a developed rostrum, which is absent in the second group [74]. *Dasyatis pastinaca*, on the other hand, showed very slow growth, both in body size with increasing age, and in the height of the vertebrae with increasing body size. In this species, the growth rate of all body parts is severely constrained by the low general metabolic rate observed in the longer-lived species that generally spend a high-energy investment in reproductive strategy [45,76,77]. It is worth noting the invariant trend of the vertebral diameter–height ratio, which highlighted a general isometry in the vertebral growth of the species studied as their body size increased. Isometric vertebral growth may be a common trait for batoid species [74], due to their typical dorso-ventrally depressed body shape with pectoral fins enlarged to varying degrees [13,19]. Despite some slight differences found between species (in the case of the common stingray, as discussed above), the incomplete sample size range (juveniles and subadults, with few adult individuals) prevented stronger evidence to support this contention.

The third point of support for the proposed method comes from the reliability of the growth parameters estimated through the VBGE and Gompertz models applied, compared with information available in the literature, although the samples suffered the consequences of a size-selective capture [24]. Indeed, they were not completely representative of the entire size range expected for the species considered, in addition to a limited and unbalanced

sample size. This problem was addressed by coupling the iteration of one of the simpler methods, the Ford–Walford method [68,69], with a bootstrap routine applied to both vertebral and body measurements.

On the one hand, the iteration of the Ford–Walford method helped to increase the accuracy of the asymptotic length estimation by providing a mean value with error. Indeed, while the Ford–Walford method suits problematic datasets (limited number of observations and lack of age groups), it provides estimates that are crude and serially correlated and depend on the sample size of the age groups [78,79]. On the other hand, bootstrapping helped account for whole species-specific variability, as the age classes considered (an intermediate portion of the expected size range) inherently contain the average information to estimate the age and growth parameters typical of the species [80,81]. Finally, the asymptotic models with length constraints worked for the interdependence of the estimate of the growth parameters, as recommended by [29], but not for the simultaneity of the latter. In the case under examination, the simultaneous estimates of asymptotic lengths k and $t_0$ would have suffered from a too-wide error range due both to the low representativeness of the older age groups [38,82] and the lack of the youngest individuals (0+ and 1 classes), i.e., a suboptimal situation for good adaptation [83]. In a similar situation, yet for a very different assemblage of species, Smart et al., 2012 [84] developed a method based on the back-calculation of length at age data to overcome the constraints of low sample sizes in age and growth studies on rare carcharhinid and hammerhead sharks. Differently from the present sample, large vertebral sizes and well-defined age rings observed in these species allowed for the successful application of back-calculation techniques to increase the accuracy in estimating their age and growth parameters. In the present study, small centra and poorly calcified age rings of the batoid species under consideration would have impeded a reliable application of such a statistical technique to assess the age and growth parameters of samples having uneven and/or lacking size classes. In the present work, this problem was overpassed by using the same samples first to calculate a set of mean asymptotic lengths and then to estimate k, $k_G$, $t_0$, and $c_G$ through the asymptotic length-constrained VBGE and Gompertz models fitting the corresponding data. Estimation of growth parameters based on a set of mean asymptotic lengths between two model types enabled cross-variable and cross-model comparisons. Comparisons between the body variables and vertebral measures illustrated by the former were better than the latter at estimating growth parameters with increasing age. In fact, the values of the rate of increment in vertebral dimension with increasing size do not always have the same rank across species compared to the rate of increment in body measures with increasing age, as demonstrated in this study. Additionally, model comparisons confirmed the Gompertz model as the best function for batoid species [85,86]. A notable exception was *T. torpedo*, which showed good adaptations despite the model and/or variable used. This suggested that different measures and logistic functions can be used for this species for age and growth studies.

The resultant estimated growth rates k, as well as the asymptotic lengths, were generally in line with data available in the literature, and this is true for three of the species studied. For example, age data for *D. pastinaca* showed growth rate and mean asymptotic DW very close to data from Turkish [57] and Greek [58] waters or higher and lower, respectively, than the coastal waters of the Egyptian Mediterranean [59]. Large individuals are rare [87,88] and other authors reported a maximum disc width of 150 cm for this species [44], which is about 50 cm greater than the asymptotic DW calculated in this study. The calculated age–growth rate for *T. torpedo* was consistent with other results on the age–weight–growth rate of the species from Spanish Mediterranean waters [60] or higher than an outdated study from the Gulf of Tunis [61]. The species is reported at the maximum size of 40 cm TL in the southern Mediterranean [88], which is about 4 cm smaller than the one estimated here in asymptotic TL. Age data for *R. asterias* showed that the actual sample grows almost as fast as a sample from a more northern area close to where it is today [62]. On the other hand, the species appears to increase in weight and size in the northwestern

Mediterranean, although the growth rate with age is not provided [41]. Estimation of age–growth parameters required caution for *T. marmorata*, due to limited sample size and sex composition (males only). In fact, while the asymptotic estimates appeared particularly lower in the marbled electric ray compared to the common torpedo, it is known that the former grows larger than the latter. The marbled electric ray grows up to 60 cm TL in the southern Mediterranean [52,88] or even up to 100 cm as the maximum recorded TL [53]. In fact, it exhibits a lower age and growth rate in Sardinian waters [63] than what was observed for the congener in this study. Even though the bootstrap routine used in the present experiment functioned on difficult samples, the case of the marbled electric ray indicates that the sample size cannot be too small and/or size–sex unbalanced in order to obtain reliable age and growth estimates by using this method.

Although data provided in this work are not validated [89], a condition that is achieved quite rarely [4], age and growth parameters are comparable across species inhabiting the same area. In fact, all estimates were obtained using both a single band pair enhancing method and statistical routine on samples obtained from a local size-selective fishery. This greatly reduced the biases connected with comparing samples with differences in statistical treatment and/or laboratory procedures of the aging methods used for different species. For instance, present data highlighted how the species-specific differences in the life history traits of the species studied can influence their age and growth rates. Indeed, the oviparous and more fecund starry skate showed the highest age and growth rate compared to the aplacental viviparous common torpedo and, above all, to the common stingrays, a large-sized species that even evolved intrauterine trophic connections with its embryos [45,76,77,90]. Comparability of the data is equally important when studying populations of elasmobranch species distributed in different areas. In this case, intraspecific variation in the age and growth parameters from comparable data can inform on differences between factors that actually influence the species growth rate, such as biogeographic and bioecological conditions (trophic conditions, endemism, home range, and gene flow, depending on the species) and in the fishing pressure exerted on populations in different areas [91].

The impact of small-scale coastal fisheries on elasmobranch-dependent coastal species is a critical issue on a global scale [92–96], especially where the elasmobranch fauna is rich and composed of large endangered species [97]. In fact, the IUCN Red List highlights how conservation measures are urgently needed for all dependent coastal species [98,99], in addition to obtaining local demographic assessments of populations under fishing pressure. In this context, studies on aging are essential to distinguish how the threat of fisheries affects populations, together with comparable demographic information that is critical for their conservation [4]. This information may be useful for addressing conservation measures for those age groups that are most affected by fishing within populations of elasmobranchs under fishing pressure [8]. For example, restrictions on fishing in too shallow waters could help limit the bycatch of the two to three-year-old juveniles and subadults that were the most represented individuals in the elasmobranch bycatch of the study area, as well as in similar areas, which are important feeding, breeding, and/or nursery areas for many coastal elasmobranchs [95,96]. As in this study, providing comparable demographic information for these species in an easily replicable method will help fill such an information gap and help conserve them.

**Supplementary Materials:** The following supporting information can be downloaded at: https://www.mdpi.com/article/10.3390/d16050271/s1, Figure S1: instrumental details of stereoscope used for age readings and measurements.

**Author Contributions:** Conceptualization, U.S.; methodology, U.S. and F.Z.; software, U.S. and F.Z.; validation, U.S. and F.Z.; formal analysis, U.S. and S.K.; investigation, U.S. and S.K.; resources, G.N., U.S. and F.Z.; data curation, U.S., E.M. and F.T.; writing—original draft preparation, U.S.; writing—review and editing, U.S., F.Z., S.K., E.M., F.T. and G.N.; visualization, U.S.; supervision,

U.S.; project administration, G.N.; funding acquisition, G.N. All authors have read and agreed to the published version of the manuscript.

**Funding:** This research received no external funding.

**Institutional Review Board Statement:** This study implemented the responsible collection of specimens sampled from the ordinary bycatch of a small-scale local fishery. Collected samples were always individuals who died at the catch and/or during onboard disentangling operations. We specifically instructed and encouraged fishermen to facilitate the release at sea of live-captured individuals when conditions were safe due to do so, due the potential dangers of handling three out of the four species under study. Specimen collection was carried out in full accordance with Italian laws and regulations in this context and conducted under the responsibility of the University of Tuscia, within a collaboration project between the University and the fishery coop of Fiora's Port Canal.

**Data Availability Statement:** The datasets generated during and/or analyzed during the current study are available from the corresponding author upon reasonable request.

**Acknowledgments:** We are grateful to the anonymous reviewers who reviewed the manuscript. We are grateful to A. Annunziatellis who provided us with the map and to all fishermen of Fiora's Port Canal who collaborated with this study.

**Conflicts of Interest:** The authors declare no conflicts of interest.

## Appendix A

**Table A1.** Regression parameters (Rp, a: slope, b: intercept, $R^2$: explained variance, Std. err: standard error, t (n − 2): t statistic for n − 2 degrees of freedom with n as sample size, and p: level of significance) of linear relationships between vertebral dimensions and increasing fish size as total length (mm) of three coastal batoid species (*RA*: *Raja asterias*, *TM*: *Torpedo marmorata*, and *TT*: *Torpedo torpedo*) sampled in the central Tyrrhenian Sea. VD, VH, and VD/VH are vertebral diameter, height, and their ratio, respectively. The same letter in brackets indicates a significantly different regression slope between species with asterisks denoting the p-level of the significance of the difference in slope in pairwise comparisons between species. ns stands for not significant differences. * $p < 0.05$, ** $p < 0.01$, *** $p < 0.001$.

| Variables | Rp | Species | | |
| --- | --- | --- | --- | --- |
| | | *RA* | *TM* | *TT* |
| VD | *a* | $8.0 \times 10^{-3}$ | $3.0 \times 10^{-3}$ | $1.4 \times 10^{-2}$ |
| | *Std. err* | $5.1 \times 10^{-4}$ | $3.4 \times 10^{-3}$ | $5.7 \times 10^{-4}$ |
| | *t (n − 2)* | 15.7 | 0.9 | 24.7 |
| | *p* | *** (a *) | ns | *** (a *) |
| | *b* | $2.0 \times 10^{-6}$ | 2.79 | −0.40 |
| | *Std. err* | $2.1 \times 10^{-1}$ | 0.96 | 0.16 |
| | *t (n − 2)* | $1.0 \times 10^{-4}$ | 2.9 | 2.4 |
| | *p* | ns | *** | * |
| | $R^2$ | 0.83 | $5.6 \times 10^{-2}$ | 0.92 |
| VH | *a* | $7.7 \times 10^{-3}$ | $4.5 \times 10^{-3}$ | $1.0 \times 10^{-2}$ |
| | *Std. err* | $6.7 \times 10^{-4}$ | $3.6 \times 10^{-3}$ | $4.5 \times 10^{-4}$ |
| | *t (n − 2)* | 11.3 | 1.3 | 22.0 |
| | *p* | *** (b *) | ns | *** (b *) |
| | *b* | −0.33 | 1.45 | −0.32 |
| | *Std. err* | 0.29 | 1.0 | 0.13 |
| | *t (n − 2)* | −1.2 | 1.4 | −2.4 |
| | *p* | ns | ns | * |
| | $R^2$ | 0.71 | 0.11 | 0.90 |

**Table A1.** *Cont.*

| Variables | Rp | Species | | |
|---|---|---|---|---|
| | | *RA* | *TM* | *TT* |
| VD/VH | *a* | $-4.4 \times 10^{-5}$ | $-1.0 \times 10^{-3}$ | $-1.0 \times 10^{-5}$ |
| | *Std. err* | $3.1 \times 10^{-4}$ | $1.5 \times 10^{-3}$ | $2.6 \times 10^{-4}$ |
| | *t (n − 2)* | $-1.4 \times 10^{-1}$ | $6.6 \times 10^{-1}$ | $-4.0 \times 10^{-2}$ |
| | *p* | ns | ns | ns |
| | *b* | 1.18 | 1.65 | 1.42 |
| | *Std. err* | 0.13 | 0.43 | $7.7 \times 10^{-2}$ |
| | *t (n − 2)* | 9.3 | 3.8 | 18.4 |
| | *p* | *** | ** | *** |
| | $R^2$ | $3.9 \times 10^{-4}$ | $3.3 \times 10^{-2}$ | $3.0 \times 10^{-5}$ |

**Appendix B**

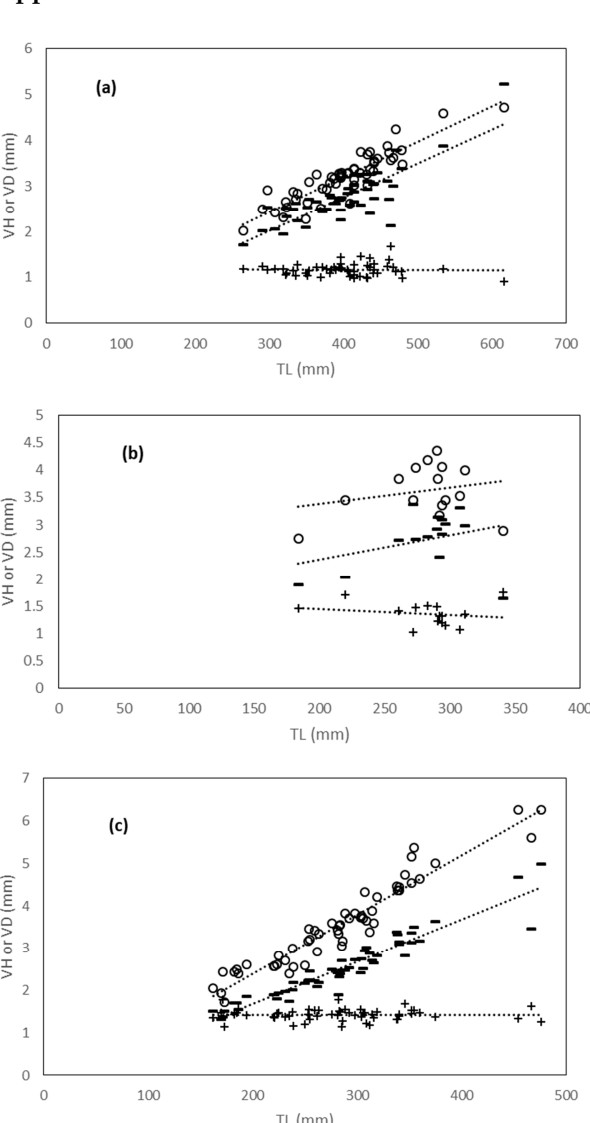

**Figure A1.** (**a–c**) Linear relationships between total length (TL) and vertebral measures (VD vertebral diameter, empty circles; VH vertebral height, minus; and VD/VH diameter/height ratio, dark crosses) in three coastal batoid elasmobranch species form the central Tyrrhenian Sea. (**a**) *Raja Asterias*; (**b**) *Torpedo marmorata*; and (**c**) *Torpedo torpedo*.

**Appendix C**

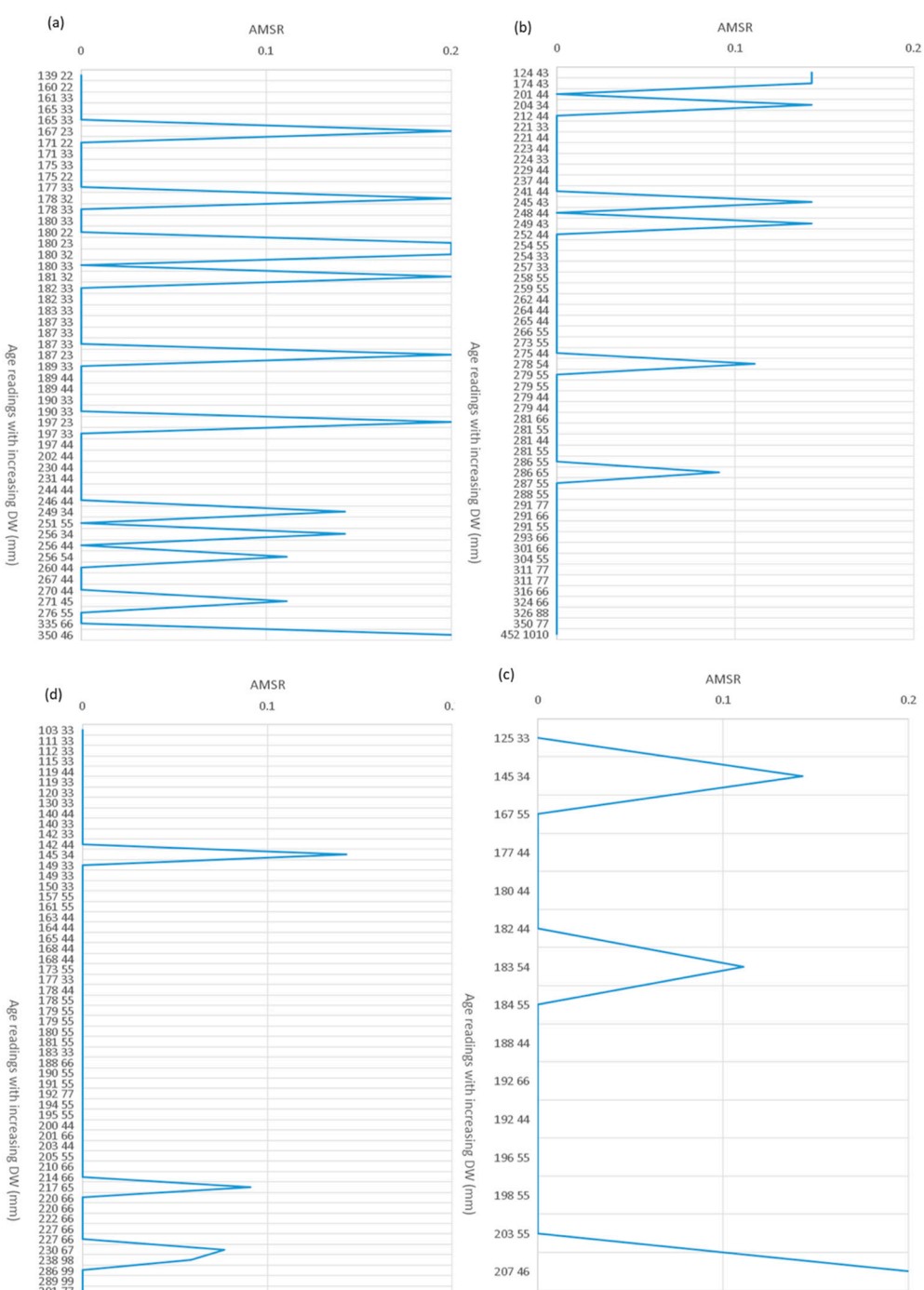

**Figure A2.** (**a**–**d**) Age bias plots representing the variation in the Absolute values of Mean-Standardized Residuals (AMSRs) between age readings (as couples of numbers placed on the right of the size in cm) of two independent readers with increasing body size as disk width (DW, mm) of four coastal batoid species sampled in the central Tyrrhenian Sea: (**a**) *Dasyatis pastinaca*; (**b**) *Raja asterias*; (**c**) *Torpedo marmorata;* and (**d**) *Torpedo torpedo*.

**Appendix D**

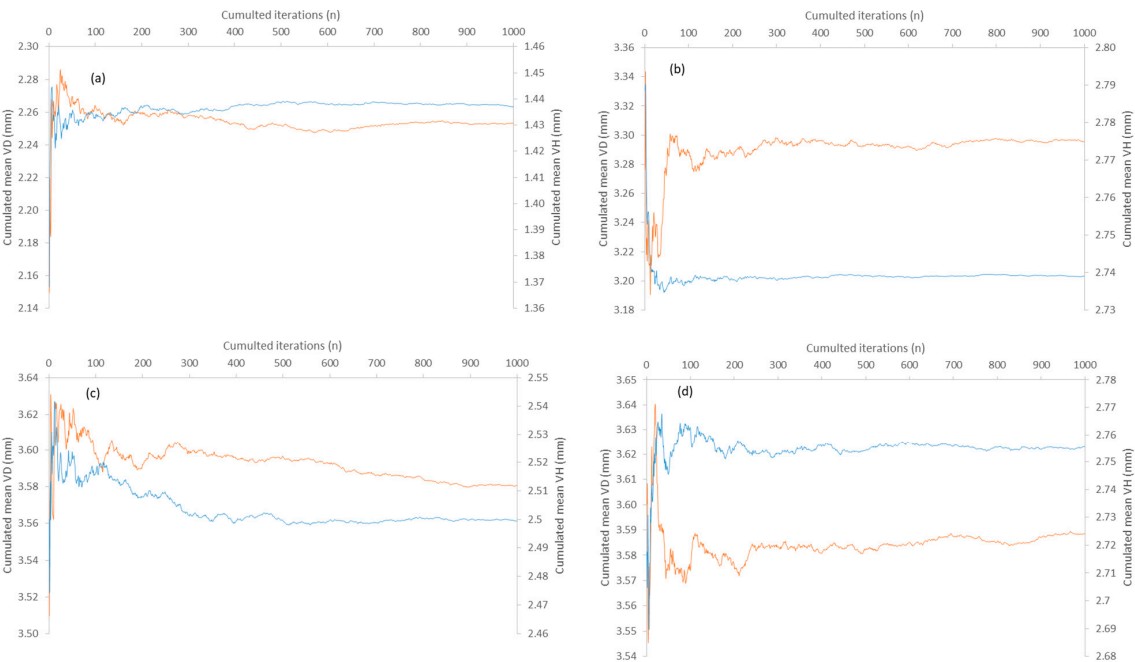

**Figure A3.** (**a**–**d**) A number of cumulated iterations are needed for stabilizing the cumulated sample mean of vertebral height (VH) orange and diameter (VD) blue in the bootstrapping procedure applied to (**a**) *Dasyatis pastinaca*, (**b**) *Raja asterias*, (**c**) *Torpedo torpedo,* and (**d**) *Torpedo marmorata* sampled in coastal waters of the central Tyrrhenian Sea.

**Appendix E**

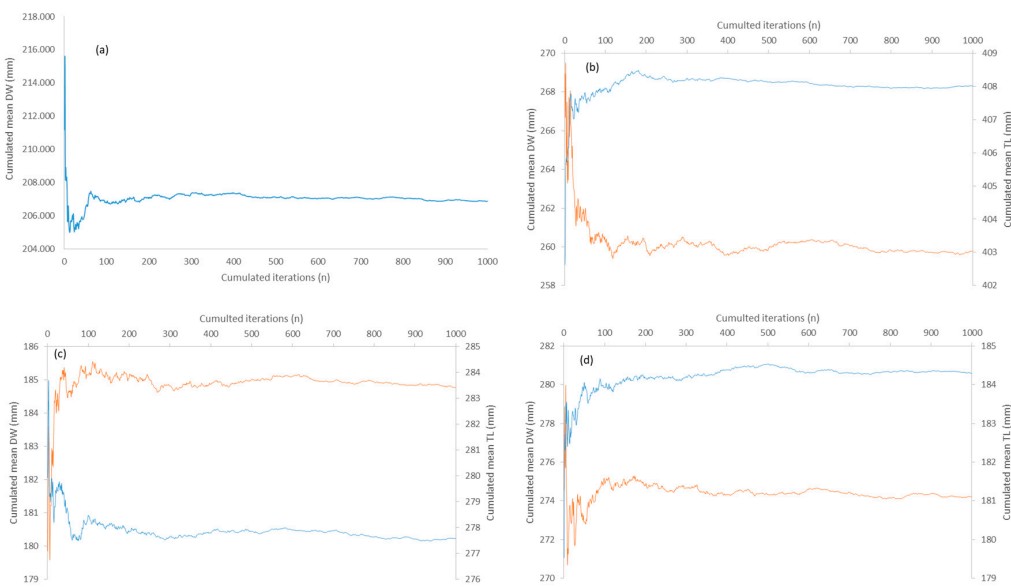

**Figure A4.** (**a**–**d**) Number of cumulated iterations needed for stabilizing cumulated sample mean of disk width (DW) orange and total length (TL) blue in the bootstrapping procedure applied to (**a**) *Dasyatis pastinaca*, (**b**) *Raja asterias*, (**c**) *Torpedo torpedo,* and (**d**) *Torpedo marmorata* sampled in coastal waters of the central Tyrrhenian Sea.

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
