# Peer review of "Age Readings and Assessment in Coastal Batoid Elasmobranchs from Small-Scale Size-Selective Fishery: The Importance of Data Comparability in Multi-Specific Assemblages"

_diversity, doi:10.3390/d16050271_

Round 1
Reviewer 1 Report
Comments and Suggestions for Authors
The article has many flaws in relation to studies of age and growth in elasmobranchs. The two requirements for carrying out studies of this type were not met. The values of the coefficients of determination (r2) are very low, demonstrating that there is not a good proportionality between the size of the vertebrae and the lengths of the individuals. The periodicity of the formation of age rings was also not estimated, and it was shamelessly considered that each ring corresponds to one year. The growth curves were estimated with a few individuals, making it necessary to use back-calculated lengths for a better adjustment, as described by Smart et al. (2012). These corrections are extremely necessary for the manuscript to be accepted by a journal.
Smart JJ, Harry AV, Tobin AJ, Simpfendorfer CA (2012) Overcoming the constraints of low sample sizes to produce age and growth data for rare or threatened sharks. Aquat Conserv Mar Freshw Ecosyst. https://doi.org/10.1002/aqc.2274
Comments on the Quality of English LanguageVery well written article in English
Author Response
Comments and Suggestions for Authors
The article has many flaws in relation to studies of age and growth in elasmobranchs. The two requirements for carrying out studies of this type were not met. The values of the coefficients of determination (r2) are very low, demonstrating that there is not a good proportionality between the size of the vertebrae and the lengths of the individuals. The periodicity of the formation of age rings was also not estimated, and it was shamelessly considered that each ring corresponds to one year. The growth curves were estimated with a few individuals, making it necessary to use back-calculated lengths for a better adjustment, as described by Smart et al. (2012). These corrections are extremely necessary for the manuscript to be accepted by a journal.
Smart JJ, Harry AV, Tobin AJ, Simpfendorfer CA (2012) Overcoming the constraints of low sample sizes to produce age and growth data for rare or threatened sharks. Aquat Conserv Mar Freshw Ecosyst. https://doi.org/10.1002/aqc.2274
Reply to Reviewer 1
We thank very much the referee for her/his comments on the ms. Nevertheless, we do not agree with the two of the main criticisms provided by Reviewer 1. Changes in the revised text are highlighted in light blue for this Referee.
1) The first criticism about R2 coefficients of the linear fits between vertebral measures (vertebral height and width) and body size (disc width and total length of the animals) is true only for one species, T. marmorata, which, indeed, was excluded from estimation of age and growth parameters due to too low sample size and unbalanced representation of sexes (only males in the sample). The results for this species were reported in the MS as a term of comparison with other species, suggesting that the method we propose cannot be applied to samples having too low a number of individuals and/or being particularly unbalanced in sex composition. This issue was stressed specifically in the submitted ms, particularly ”Even though the bootstrap routine used in present experiment functioned on difficult samples, the case of the marlbed electric ray indicated sample size cannot be too small and/or size-sex unbalanced for obtaining reliable age and growth estimates by using this method.” (lines 684 - 687, page 20, discussion section in the revised version). For all other species investigated, values of R2 in linear models were acceptably high (Zar, 1999) as such values ranged between the 61% and 92% of explained samples ‘variance (in particular: 0.83 and 0.92 in VD-TL models, and 0.71 and 0.90 in VH-TL models, for R. asterias and T. torpedo, respectively, see supplemental table in Appendix A1; 0.71, 0.72 and 0.85 in VD-DW models, and 0.61, 0.70 and 0.81 in VH-DW models, for D. pastinaca, R. asterias and T. torpedo, respectively, see table 2 for details). Confirming these data, estimates of regression slope (parameter a) had statistically significant values in the linear models cited above for the species considered (see table 2 and supplemental table in Appendix A1 for details). Therefore, the first requirement for data verification in age and growth studies was met in the species which age and growth parameters were estimated for.
2) The assessment of the periodicity of age ring formation, i.e. the validation of age and growth data, was not the aim of this study, and this was clearly stated in the submitted ms, in particular “Although data provided in this work are not validated [90], a condition that is achieved quite rarely,..." (lines 688 - 690, page 20 in the discussion section in the revised version). Age and growth data validation requires other investigations, for instance a sequence of catch, tetracycline treatment, tagging, and recapture data of samples after a representative period of time. However, we noticed that we made mistakenly, and unintentionally, some reference to annual deposition in the ring formation process in the result section and related graphs. Therefore, we corrected the corresponding text as follows:
Lines: 255 - 260 , page 6: the word “year/s” was replaced with the word “count/s”
And replaced the word “years” with “counts” in figure 6 a, b, c, d and 7 a, b, c, d
3) The reference work (Smart et al., 2012) the referee suggested us to improve estimation of age and growth parameters is very interesting but we think it is not applicable to our samples for a couple of reasons. First, it was applied on a very different assemblage of species, i.e. large charcharinid and hammerhead shark species, compared to small medium-sized batoids investigated in present ms. The species considered in Smart et al 2012 have large and well calcified vertebra that allow unbiased identification of age rings and easy counting, even without using histology of some sort. Second, this has consequences on the accuracy of measurements of vertebral increments, which is mandatory for a reliable back calculation of fish length. Obtaining accurate measurements of vertebral increments observed between contiguous couples of age rings strictly requires observing vertebral sections (histological or not) that are perfectly placed orthogonally to the observer, in addition to well defined age rings on large centra. This was not the condition we found in present sample, as it shows very small centra that complicated the identification of age rings. In addition, even though samples were placed orthogonally to the observers as best as we can during image acquisition, we cannot rule out the possibility that sample orthogonality was not perfect during photo recording. Such a source of bias would have been greater when measuring increments between couples of contiguous age rings compared to assessing the whole vertebra height and width. However, we added text acknowledging and discussing the important reference by Smart et al., 2012. In particular,
Lines 640 - 648, page 19 : “In a similar situation, yet for a very different assemblage of species, Smart et al 2012 [84] developed a method based on back calculation of length at age data to overcome the constraints of low sample sizes in age and growth studies on rare carcharhinid and ham-merhead sharks. Differently from present sample, large vertebral size and well defined age rings observed in these species allowed successfully applying back calculation techniques to increase the accuracy in estimating their age and growth parameters. In present study, small centra and poorly calcified age rings of the batoid species under consideration would have impeded a reliable application of such a statistical technique to assess age and growth parameters of samples having uneven and/or lacking size classes.”
Comments on the Quality of English Language
Very well written article in English
Reviewer 2 Report
Comments and Suggestions for Authors
Abstract:
The abstract is very general, there’s very little information on the data acquired in the work. The acquired information and its discussion is of greater value than the replicability of the method used.
Introduction:
The introduction is relatively short, with insufficient information concerning different methods/techniques that can be used for acquiring the desired results. Information on the four batoid species used is also lacking. Please expand the introduction to cover these points.
Materials and Methods:
In section 2.3. Laboratory preparation of vertebrae, as elsewhere in the text, it is preferable to use passive mode of the verb instead of the noun „we“ noun, followed by the verb. Please revise here and elsewhere in the text.
Gliceryne formula is superfluous (it is a simple and common compound, its structure is widely known).
In section 2.4. Statistical methods, the
Listing the commands used and steps taken during statistical analysis is unnecessary, I would advise to remove them (rows 224 and 225).
Comments on the Quality of English LanguageThe quality of work in England is good. But the address and narrative in the work must be improved, because the word we is used too much when describing the work and observations.
Author Response
Comments and Suggestions for Authors
Abstract:
The abstract is very general, there’s very little information on the data acquired in the work. The acquired information and its discussion is of greater value than the replicability of the method used.
We thank the referee for this comment. We added information required in the abstract section. Changes in the revised text are highlighted in yellow for this Referee. In particular:
- the number of sampled individuals
- fishing depth and period of the sampling
- level of acceptable disagreement within and between reders
- value ranges of coefficient of growth in size with increasing age in three of the species studied
Introduction:
The introduction is relatively short, with insufficient information concerning different methods/techniques that can be used for acquiring the desired results. Information on the four batoid species used is also lacking. Please expand the introduction to cover these points.
We added more information on the general methods used for reading age in elasmobranch species in the introduction section. In doing this task, we also added some references regarding age and growth studies in the species here considered. In particular:
Lines 43 - 52, pages 1 -2 “Most ageing techniques involve staining procedures with expensive reagents (e.g., alizarin red, hematoxylin, xylene, crystal violet, graphite microtopography, cobalt nitrate, ammonium sulfide, Safranin O, Alcian blue, copper lead, iron-based salts, and silver nitrate), the operation of which depends strictly on the species being analyzed [4; 9; 15; 61 - 64]. This is particularly true in the vertebrae of batoid species, as their vertebrae are difficult to read without histology or staining of some sort in many species [12]. On the other hand, the vertebrae of coastal species, such as carcharhinids and lamnids, do not require any stain techniques and can be read directly [12]. The most advanced use spectroscopy, from near-infrared spectroscopy to X-ray techniques [4; 15; 65 -67].”
Lines 97 - 104, pages 2 -3 “Age and growth studies in the Mediterranean Sea are generally scanty and/or dated for the species here considered, such as works carried out for D. pastinaca in the Turkish [81], Greek [82] the Egyptian [83] waters, for T. torpedo in Spanish waters [85] and in the Gulf of Tunis [86], for R. asterias in the northern Tyrrhenian Sea [88] and northwestern Mediterra-nean [35], and for T. marmorata in Sardinian waters [89]. Based on these data, the common stingray and the starry skate exhibit the lowest and the highest growth rate, respectively, among the species considered, with the marbled electric ray having a slower growth compared to the common torpedo.”
Materials and Methods:
In section 2.3. Laboratory preparation of vertebrae, as elsewhere in the text, it is preferable to use passive mode of the verb instead of the noun „we“ noun, followed by the verb. Please revise here and elsewhere in the text.
We revised the text according to the referee’s suggestion by using mainly the passive mode in phrasing throughout the ms
Gliceryne formula is superfluous (it is a simple and common compound, its structure is widely known).
We revised the text according to the referee’s suggestion and removed the glycerine chemical formula
In section 2.4. Statistical methods, the Listing the commands used and steps taken during statistical analysis is unnecessary, I would advise to remove them (rows 224 and 225).
We revised the text according to the referee’s suggestion and removed the excel line command
Comments on the Quality of English Language
The quality of work in England is good. But the address and narrative in the work must be improved, because the word we is used too much when describing the work and observations.
We revised the whole ms by using mainly the passive mode in phrasing.
Reviewer 3 Report
Comments and Suggestions for Authors
This manuscript describes a method that can be used across elasmobranch species to estimate age with relatively straightforward and simple techniques. The usefulness is clear for smaller individuals, less so with larger although the fisheries under investigation were size selective, so that is not necessarily a criticism. Use of this method to identify bycatch impacts on nursery areas and other essential habitats should be emphasized in the Abstract (as well as the text) as a potential positive outcome of this study.
Appendix Fig. A3 belongs directly in the text in my opinion, and perhaps current Fig. 3 could be moved to the Appendix.
Author Response
Comments and Suggestions for Authors
This manuscript describes a method that can be used across elasmobranch species to estimate age with relatively straightforward and simple techniques. The usefulness is clear for smaller individuals, less so with larger although the fisheries under investigation were size selective, so that is not necessarily a criticism. Use of this method to identify bycatch impacts on nursery areas and other essential habitats should be emphasized in the Abstract (as well as the text) as a potential positive outcome of this study.
We added a final statement in the abstract regarding bycatch impacts on nursery areas and other essential habitat. In the final part of the discussion section this issue was already stressed. Changes in the revised text are highlighted in green for this Referee
Lines 37 - 38, page 1 “…in potential nursery areas and other essential habitats for elasmobranchs.”
Appendix Fig. A3 belongs directly in the text in my opinion, and perhaps current Fig. 3 could be moved to the Appendix.
We exchanged the two figures according to the referee’s comment and renumbered the figures accordingly.